# Brazilian Populations of *Aedes aegypti* Resistant to Pyriproxyfen Exhibit Lower Susceptibility to Infection with Zika Virus

**DOI:** 10.3390/v14102198

**Published:** 2022-10-06

**Authors:** Kauara Brito Campos, Abdullah A. Alomar, Bradley H. Eastmond, Marcos Takashi Obara, Barry W. Alto

**Affiliations:** 1Entomology and Nematology Department, Florida Medical Entomology Laboratory, Institute of Food and Agricultural Sciences, University of Florida, 200 9th SE, Vero Beach, FL 32962, USA; 2Laboratório de Parasitologia Médica e Biologia de Vetores, Faculdade de Medicina, Universidade de Brasília, Campus Universitário Darcy Ribeiro, Asa Norte, Brasília 70910-900, Brazil; 3Coordenação Geral de Vigilância de Aboviroses, Secretaria de Vigilância em Saúde, Ministério da Saúde, Edifício PO 700, SRTV 702, Via W 5 Norte, Brasília 70723-040, Brazil

**Keywords:** Zika virus, *Aedes aegypti*, pyriproxyfen, per os infection, insecticide resistance, viral titer, vector competence

## Abstract

Zika virus (ZIKV) infection has caused devastating consequences in Brazil as infections were associated with neurological complications in neonates. *Aedes aegypti* is the primary vector of ZIKV, and the evolution of insecticide resistance (IR) in this species can compromise control efforts. Although relative levels of phenotypic IR in mosquitoes can change considerably over time, its influence on vector competence for arboviruses is unclear. Pyriproxyfen (PPF)-resistant populations of *Ae. aegypti* were collected from five municipalities located in Northeast of Brazil, which demonstrated different resistance levels; low (Serrinha, Brumado), moderate (Juazeiro do Norte, Itabuna), and high (Quixadá). Experimental per os infection using ZIKV were performed with individuals from these populations and with an insecticide susceptible strain (Rockefeller) to determine their relative vector competence for ZIKV. Although all populations were competent to transmit ZIKV, mosquitoes derived from populations with moderate to high levels of IR exhibited similar or lower susceptibility to ZIKV infection than those from populations with low IR or the susceptible strain. These observations suggest an association between IR and arbovirus infection, which may be attributable to genetic hitchhiking. The use of PPF to control Brazilian *Ae. aegypti* may be associated with an indirect benefit of reduced susceptibility to infection, but no changes in disseminated infection and transmission of ZIKV among PPF-resistant phenotypes.

## 1. Introduction

Zika virus (ZIKV) is an arthropod-borne virus (arbovirus) belonging to the family *Flaviviridae* and genus *Flavivirus*. It was first isolated from a rhesus sentinel monkey caged in the Zika forest of Uganda in 1947 and then from Aedes (*Stegomyia*) africanus mosquitoes in the same forest in 1948 [1]. Human infection was reported in Nigeria in 1953 [2]. In the following years, evidence of antibodies to ZIKV were found in human populations throughout Sub-Saharan Africa and Southeast Asia, despite the occurrence of a few symptomatic cases [3]. Zika virus is primarily transmitted by mosquito vectors, mainly attributed to the subgenus *Stegomyia* of the genus *Aedes* [3], where *Ae. aegypti* is considered as the primary ZIKV vector to humans [4]. There is evidence showing that ZIKV can be spread to humans by different means of transmission, including breastfeeding [5], sexual [6], and blood transfusion [3], which may complicate the control of this virus. Infections of ZIKV in humans are often asymptomatic; when present, symptoms include mild fever, maculopapular rash, myalgia, arthralgia, retro-orbital pain, conjunctivitis, and headache [7].

Zika virus was not considered as a public health problem for decades due to few human clinical cases [8]. However, the first outbreak of ZIKV occurred in 2007 on Yap Island, Micronesia, caused by the Asian ZIKV lineage. Approximately, 73% of island residents were infected and reported mild and short-lived symptoms [8]. From 2013 to 2014, French Polynesia experienced an outbreak of ZIKV with high attack rates and cases associated with neurological complications (Guillain-Barré syndrome) [9]. The virus spread throughout the South Pacific in 2014, provoking outbreaks in New Caledonia, the Cook Islands, and Easter Island [10]. In November 2014, states in Brazil’s Northeast region reported outbreaks of ZIKV infection [11], which were laboratory confirmed the following year [12]. The French Polynesian ZIKV strain, the Asian lineage of the virus, was speculated to have entered Brazil between May and December 2013 during specific sporting events [13]. In 2015, it has been estimated that there were between 440,000 and 1,300,000 cases of ZIKV in Brazil. [14]. The association between ZIKV infection and neurological signs/symptoms in adults has been confirmed [15], followed by confirmation of the association between ZIKV infection and neonatal microcephaly after the virus isolation from amniotic fluid and fetal brain tissue [16]. Retrospective analyses suggested that births of babies with microcephaly were associated with ZIKV infection and also occurred during the outbreak in French Polynesia, in 2013 and 2014 [17]. Later, autochthonous transmission of ZIKV has spread to all Brazilian states [18], the South and Central Americas, and the Caribbean [19].

Pyriproxyfen (PPF) is an insect growth regulator that targets mosquito immature stages, disrupts their development, and inhibits their emergence to adulthood [20,21,22]. Pyriproxyfen has been used to control *Ae. aegypti* in public health throughout Brazil since 2014, largely attributable to insecticide resistance (IR) in *Ae. aegypti* to other insecticides, including pyrethroids and organophosphates [23,24]. The most studied mechanisms responsible for the evolution of resistance are increased activity of detoxifying enzymes and modification of the insecticide target site. The resistance may have a cost on insects’ biology as it can affect their life history traits, such as larval development, sexual competition, susceptibility to predation, and vector competence [25,26,27,28,29]. Assessment of the susceptibility status of 123 *Ae. aegypti* populations collected throughout Brazil between 2017 and 2018 detected IR to PPF in populations from municipalities in the states of Bahia and Ceará, Northeast region, for the first time in Brazil [24]. It is known that IR can affect vector control efforts and epidemic prevention; however, few studies have assessed the influence of IR on *Ae. aegypti* vector competence for arboviruses. To our knowledge, no study has been published evaluating the association between PPF resistance and vector competence of arboviruses. New collections of *Ae. aegypti* populations were carried out about 2 to 3 years later from previous municipalities (Serrinha, Brumado, Juazeiro do Norte, Itabuna, Quixadá) to investigate additional changes in resistance levels following continued PPF pressure. Here, we test the hypothesis that vector competence in Brazilian populations of *Ae. aeygpti* varies with PPF resistance level.

## 2. Materials and Methods

### 2.1. Ethics Statement

Mosquito infection experiments with ZIKV were performed in an arbovirology research facility (Biosafety level-2 and Arthropod Containment Level-2) at the Florida Medical Entomology Laboratory in accordance with the approved protocol by the University of Florida’s Institutional Biosafety Committee and Animal Care and Use Committee.

### 2.2. Mosquito Populations

*Aedes aegypti* mosquito populations were collected in February and March 2020 from different cities (Serrinha, Brumado, and Itabuna, from Bahia state; Juazeiro do Norte, and Quixadá, from Ceará state) located in Northeast Brazil (Figure 1) using 100 ovitraps in towns with up to 50,000 houses and 150 ovitraps in cities with up to 200,000 homes, following the MoReNAa Network methodology [30,31]. These ovitraps containing tap water, yeast extract solution (0.04%), and a single wooden paddle were distributed on the grounds of selected houses in a grid pattern for 15 days, covering the urban territory to include regions presenting different infestations levels. Field-collected parental generation eggs were hatched and used to establish laboratory colonies of each population, which were maintained at the Laboratory of Physiology and Control of Arthropod Vector (Laboratório de Fisiologia e Controle de Artrópodes Vetores, LAFICAVE), at the Oswaldo Cruz Institute (IOC/Fiocruz), Rio de Janeiro/RJ before being shipped to Florida Medical Entomology Laboratory, Vero Beach.

Mosquitoes were maintained in an insectary at 26–28 °C, relative humidity of 70–80%, and a 12∶12 h light–dark cycle. Larvae were hatched in deoxygenated water prepared in an insulated vacuum container powered by an electronic pump for 45 min, and newly hatched larvae were reared in pans of distilled water (1.5 L) and fed with fish food (TetraMin). Pupae were transferred to adult rearing cages, where emergent adults were maintained on a diet of 10% sucrose solution. To generate eggs, adult females were blood-fed on restrained chickens (*Gallus gallus domesticus*) according to animal use and care policies of the University of Florida’s Institutional Animal Care and Use Committee (IACUC Protocol 202007682). Newly laid eggs were collected, kept moist for at least 24 h, and air-dried prior to storage at room temperature in plastic containers. Larval bioassays were conducted in all mosquito populations and in a Rockefeller susceptible stain (reference) to determine their resistance levels to PPF, following procedures described in the WHO guidelines for larvicidal bioassays. Modifications to the guidelines, such as changes in the number of exposed larvae, number of replicates, and amount of food offered to the larvae, were performed [32]. The results of bioassays revealed that low resistance rates in Serrinha and Brumado populations, moderate resistance in Juazeiro do Norte and Itabuna population, and high resistance in Quixadá population. The full reports of IR of these populations to PPF are submitted for publication elsewhere.

### 2.3. Zika Virus and Cells

An isolate of ZIKV (Puerto Rico strain, PRVABC59) of the Asian lineage used in this study was collected from serum of a ZIKV-infected human who traveled to Puerto Rico in 2015 and provided to us by the U.S. Centers for Disease Control and Prevention. We passaged ZIKV three times in cell culture before use in the infection study involving *Ae. aegypti* populations from Brazil. The complete genome sequence of this virus strain can be found under Gene bank accession # KU501215.1. The propagation of ZIKV was performed as previously described [33]. Briefly, African green monkey (Vero) cells were allowed to grow in T-175 cm^2^ cell culture flasks at 37 °C and 5% carbon dioxide atmosphere until 80–90% confluency. Cell monolayers were then infected with stock ZIKV at a 0.01 multiplicity of infection and incubated for six days at 37 °C in media (M199) (HyClone, Medium 199, GE Healthcare, Logan, UT, USA) supplemented with 10% heat-inactivated fetal bovine serum (Thermo Fisher Scientific, Waltham, MA, USA), antibiotics (penicillin–streptomycin), and Mycostatin. After this incubation period, infectious cell culture supernatant was harvested and added to defibrinated bovine blood (Hemostat Laboratories, Dixon, CA, USA) with adenosine-5’-triphosphate disodium salt trihydrate (ATP, Thermo Fisher Scientific, Waltham, MA, USA) to obtain infectious bloodmeal [34,35].

### 2.4. Zika Virus-Infected Blood and per os Infection

Female mosquitoes aged 5–8 days old were kept in plastic cups and deprived of sucrose solution 24 h before per os infection challenges on viremic bloodmeals administered via Hemotek feeders (Discovery Workshops, Lancashire, UK) covered with sheep intestine as a membrane and heated to 37 °C. Mosquitoes were allowed to feed for 45 min at 28 °C. Immediately after feeding, infectious bloodmeals were aliquoted into cryovials from Hemotek feeders and frozen at −80 °C for later assays to determine viral titers fed by mosquitoes. Fully engorged females were separated from partial and unfed individuals using anesthetization with carbon dioxide and transferred to new plastic cups with access to 10% sucrose solution for the duration of the experiment. The vector competence study was performed with three biological replicates.

### 2.5. Cationic-(Q)-Paper and Saliva Collection

To examine viral transmission efficiency, the saliva of females was collected using the cationic-(Q)-paper (CQP) method as described elsewhere [33]. Briefly, after an incubation period of 14 days post-infection (dpi), mosquitoes were transferred individually to *Drosophila* cultivation vials (Thermo Fisher Scientific, Waltham, MA) containing wet CQPs treated with honey that were placed on the mesh at the top of the cultivation vials to collect their saliva upon feeding on honey-containing CQPs. In order to determine whether a female fed on honey and deposited saliva, blue food coloring was mixed with honey and used as a visual marker. Mosquitoes were kept in the cultivation vials and allowed to feed on the honey-containing CQPs for 24 h. Following feeding, females were anesthetized with carbon dioxide and dissected to separate their bodies from legs, andeach mosquito sample (body, leg, CQP) was stored separately in microcentrifuge tubes (Thermo Fisher Scientific, Waltham, MA) containing M199 and frozen at −80 °C until further processing. Saliva was assayed for the presence of ZIKV from females that were successfully fed on honey-containing CQPs as indicated by visualization of blue coloring in their crops.

### 2.6. Ribonucleic Acid Extraction and Real-Time PCR for Zika Virus Quantification

Viral RNA was extracted from the frozen mosquito bodies, legs, and CQPs using QIAamp Viral RNA Mini Kit (Qiagen, Germantown, MD, USA), following homogenization of samples with steel BBs in a TissueLyser II (Qiagen, Hilden, Germany) at 19.5 Hz for 3 min and centrifuged at 13,200 rpm for 5 min. Zika virus RNA was detected and quantified by quantitative CFX96 real-time PCR system (qRT-PCR) (BioRad Laboratories, Hercules, CA). The primer sequences used were F: 5′-CTTCTTATCCACAGCCGTCTC-3′ and R: 5′-CCAGGCTTCAACGTCGTTAT-3′, with probe sequence: 5′-/56 FAM/AGAAGGAGACGAGATGCGGTACAGG/3BHQ_1/-3′. Reactions were performed using SuperScript III One-Step RT-PCR System with Platinum Taq Polymerase (Invitrogen). The conditions of the qRT-PCR were 94 °C for 2 min, and 39 cycles of 94 °C for 15 s, 50 °C for 30 min, and 58 °C for 1 min. Titers of virus in mosquito tissues and saliva were quantified using a standard curve that compares cDNA synthesis to a range of ZIKV serial dilutions in parallel with plaque assays of the same dilutions of the virus, expressed as plaque forming unit equivalents (PFUe)/mL.

### 2.7. Statistical Analysis

Susceptibility to infection (number of positive bodies/total number of mosquitoes tested), disseminated infection (number of infected legs/total number of infected bodies), and saliva infection (number of positive saliva/total number of infected legs) were calculated for each population and analyzed using logistic regression analysis (PROC LOGISTIC, SAS 9.4). Significant treatment effects were further examined using pairwise comparisons of treatment groups as follow-up tests, correcting for multiple comparisons by the Tukey–Kramer method. Viral titers in mosquito samples were tested for differences among mosquito populations using analysis of variance (ANOVA) followed by Tukey’s range test for multiple comparisons. *p*-values lower than 0.05 were considered statistically significant.

## 3. Results

To determine mosquito susceptibility to infection with ZIKV, a total of 276 females from all resistant populations that imbibed the infectious blood meal containing viral titer of 6.35 log_10_ PFUe/mL and survived the 14 days incubation period were tested for the presence of ZIKV in their bodies following viral RNA extraction. The viral titer of infectious bloodmeals approximated the viremic titers observed in humans [36].

Logistic regression analysis showed a highly significant effect of PPF resistance on mosquito susceptibility to ZIKV infection (χ^2^ = 41.50, df = 5, *p* < 0.0001). Susceptibility to infection varied among the five mosquito populations, where low resistant populations exhibited higher infection rates than moderate or high resistant populations (Table 1). Thus, the Brazilian populations of *Ae. aegypti* resistant to PPF exhibit similar or lower susceptibility to infection with ZIKV than individuals from weakly resistant and susceptible populations of *Ae. aegypti*.

Analysis of variance showed that PPF resistance had a significant effect on ZIKV titers in bodies (F = 2.33, df = 5, *p* = 0.04) and saliva (F = 2.74, df = 5, *p* = 0.04), but not in legs (F = 1.31, df = 5, *p* = 0.26). Although ZIKV prevalence of infection was highly variable between populations, only difference in viral titer was observed between the low resistant (Brumado) and moderate resistant (Juazeiro do Norte) populations (Figure 2A). Similar means of ZIKV titers were detected in legs of mosquitoes from different populations (Figure 2B). However, low resistant population (Serrinha) exhibited higher viral titers in comparison to the insecticide susceptible strain (Rockefeller) (Figure 2C).

## 4. Discussion

The Northeast region of Brazil was the first and most severely affected by the ZIKV epidemic in 2014 and 2015 [11,15]. Three years later, *Ae. aegypti* populations resistant to the larvicide PPF were detected in some cities in the same region [24], which is predicted to undermine the control of mosquito-borne arboviruses. However, some authors have suggested that IR may also impinge on mosquito control through alterations in vector traits, especially vector longevity, competence, and behavior [37]. We investigated the relationship between PPF resistance and susceptibility to ZIKV infection and transmission in *Ae. aegypti* populations collected from five cities in the Northeast region of Brazil, which demonstrated different levels of resistance to PPF: low (Serrinha, Brumado), moderate (Juazeiro do Norte, Itabuna), and high (Quixadá).

Previous studies have examined the impacts of exposure to PPF on mosquito vector competence for arboviruses [20,33]. For instance, exposing mosquitoes to PPF altered their susceptibility to ZIKV, specifically, individuals exposed to PPF at larval stages exhibit enhancement of ZIKV infection and transmission in *Ae. aegypti* [33]. The authors suggested that alteration in vector competence for ZIKV in mosquitoes surviving PPF exposure during immature stages may be associated with variation in adult size and other physiological changes (e.g., stress-induced changes) that may modify their responses to virus infection [33]. The previous studies occurred during a single generation. In contrast, assessments of the influence of IR on vector competence of arboviruses occurs over multiple generations, allowing for the possibility of the evolution of IR. Thus, differences in vector competencies may not be expected to share similar directional effects or mechanisms. To our knowledge, our study is one of the first studies to examine the influence of different levels of PPF resistance on mosquito responses to arbovirus infection. We showed that populations from Itabuna and Quixadá which exhibit moderate to high resistance levels to PPF, respectively, had lower susceptibility to ZIKV infection than other populations, including the insecticide susceptible strain (Rockfeller). Our results showed a potential association between IR and arbovirus infection, which may be attributable to genetic hitchhiking. Chromosomal linkage can allow for the possibility that dynamics of a selected site affect the genetic variation at nearby neutral loci, termed “genetic hitchhiking” [38]. Studies with *Drosophila* and *Ae. aegypti* show that genetic variation of selectively neutral loci in a genome can be constrained by fixation of advantageous mutations associated with hitchhiking effect [39]. That is, variation of selectively neutral loci may be constrained by a genetic hitchhiking effect in genome regions under extensive selection, as can be the case in which mosquitoes are frequently being exposed to insecticides [39,40]. For example, genes closely linked to the organophosphate insecticide target loci are predicted to exhibit reduced genetic variation because of a genetic hitchhiking effect associated with intensive organophosphate insecticide selection [39]. Similar concepts have been investigated and suggested for the impact of insecticide resistance on *Culex pipiens* immunity [41].

The mechanisms of PPF resistance in mosquitoes are not yet fully understood. The involvement of overexpression of P450 and GST detoxification enzyme activities was suggested [42]. Significant increases in detoxification enzyme activities were also reported in *Ae. albopictus* resistant to PPF, corroborating the hypothesis that such resistance has a metabolic basis [43]. In agreement with our findings, [44] evaluated the susceptibility of mosquitoes with different pyrethroid resistance profiles to infection and transmission of dengue viruses (DENVs), and the authors found a negative association between the frequency of *kdr* mutations (related to pyrethroid knockdown resistance) and vector competence for DENV in *Ae. aegypti*. Specifically, mosquitoes with the highest frequency of mutant alleles L1016 and C1534 had lower infection and transmission rates than those presenting low or no frequency of *kdr* mutant alleles [44]. Furthermore, [45] reported lower rate of adult lifespan in deltamethrin- resistant *Ae. albopictus* in comparison to the susceptible line, and when these two lines were orally challenged with DENV-2, individuals’ resistant to deltamethrin showed a reduction in infection rates and viral titers of DENV-2 in the head, salivary glands, and ovaries at 14 dpi. The authors also found lower horizontal transmission to mice and vertical transmission to their progeny in those resistant mosquitoes [45]. The highest deltamethrin resistance level was associated with the lowest dissemination rates of chikungunya virus in *Ae. aegypti* [46]. Together, these observations along with our study suggest that mosquitoes exhibiting resistance to insecticides may experience fitness costs and lower susceptibility to infection with arboviruses.

On the other hand, other studies observed a positive association between IR and vector competence for arboviruses. A study by [47] determined the influence of IR on lines of organophosphate-resistant *Cx. quinquefasciatus*, where these lines demonstrated higher vector competence than insecticide-susceptible reference line for West Nile virus. Higher infection rates of ZIKV were observed in a pyrethroid-resistant population of *Ae. aegypti* [48]. Permethrin-selected *Ae. aegypti* mosquitoes demonstrated significant increase in dissemination rates of DENV-1 [49]. Although the reasons responsible for these variations between IR and vector competence studies are not known, they may be attributable to different mechanisms involved in IR, genetic backgrounds between mosquito populations, and virus strains.

In conclusion, our data revealed that Brazilian *Ae. aegypti* populations with different PPF resistance profiles show altered susceptibility to ZIKV infections, specifically lower infection rates among populations of Itabuna and Quixadá which exhibited moderate and high resistance to PPF, receptively. Although our study suggested an indirect benefit of reduced infection rates of ZIKV in some PPF-resistant mosquito populations, it is essential to continue applying IR mitigation strategies considering that all populations were able to transmit ZIKV and given that previous reports detecting increases in vector competence of resistant mosquitoes for some arboviruses. Further studies are needed to better understand the relationship between IR and mosquito responses to arbovirus infection and transmission.

## Figures and Tables

**Figure 1 viruses-14-02198-f001:**
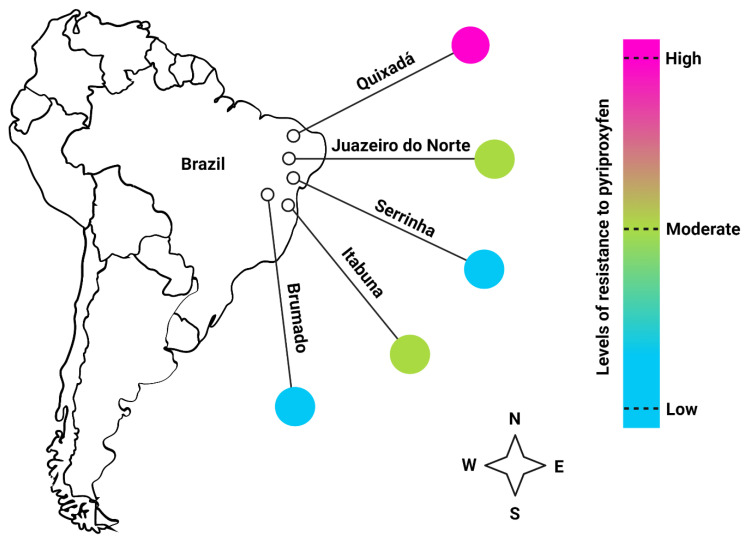
Brazil map showing cities where *Ae. aegypti* mosquito populations were collected with their relative resistance levels to PPF.

**Figure 2 viruses-14-02198-f002:**
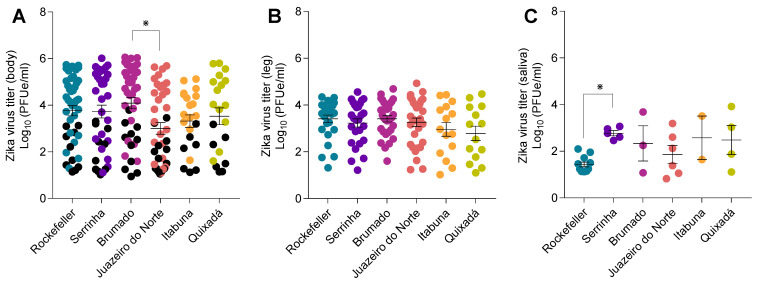
Titers of ZIKV in bodies (**A**), legs (**B**) and saliva (**C**) at 14 dpi for individuals derived from five Brazilian populations of *Ae. aegypti* with different levels of PPF resistance and an insecticide susceptible strain of *Ae. aegypti* (Rockefeller). Bars represent means  ±  standard error of the means. Black and color circles in (**A**) represent viral titers for mosquitoes with non-disseminated infections (i.e., ZIKV infection limited to midgut) and disseminated infections (ZIKV infection spread to the hemocoel), respectively. Asterisk symbols above the graphs denote significant differences following comparisons of viral titers between infected groups.

**Table 1 viruses-14-02198-t001:** Logistic regression of resistant mosquito populations on susceptibility to infection (body). Results show the means (probability scale), standard errors, and 95% confidence intervals (lower and upper means) for susceptibility to Zika infection. Means followed by different letters denote significant differences.

Population (Resistance Level)	Mean (No. Samples)	Std. Error of Mean	Lower Mean	Upper Mean
Rockefeller (reference strain)	0.8491 (51) a	0.04917	0.7262	0.9227
Serrinha (Low)	0.9512 (39) a	0.03364	0.8248	0.9878
Brumado (Low)	0.8600 (48) a	0.04907	0.7343	0.9318
Juazeiro do Norte (Moderate)	0.8200 (48) a	0.05433	0.6889	0.9036
Itabuna (Moderate)	0.4694 (47) b	0.07129	0.3354	0.6079
Quixadá (High)	0.5111 (43) b	0.07452	0.3682	0.6523

No significant effects of PPF resistance were observed on disseminated infection of ZIKV into leg tissue (χ^2^ = 3.37, df = 5, *p* = 0.64), or transmission by infected saliva (χ^2^ = 8.25, df = 5, *p* = 0.14, Table 2 and Table 3).

**Table 2 viruses-14-02198-t002:** Logistic regression of resistant mosquito populations on susceptibility to disseminated infection (legs). Results show the means (probability scale), standard errors, and 95% confidence intervals (lower and upper means) for disseminated Zika infection. Means followed by different letters denote significant differences.

Population (Resistance Level)	Mean (No. Samples)	Std. Error of Mean	Lower Mean	Upper Mean
Rockefeller (reference strain)	0.5869 (44) a	0.07260	0.4414	0.7187
Serrinha (Low)	0.6750 (38) a	0.07406	0.5173	0.8010
Brumado (Low)	0.7273 (42) a	0.06714	0.5787	0.8381
Juazeiro do Norte (Moderate)	0.7561 (40) a	0.06707	0.6031	0.8634
Itabuna (Moderate)	0.6667 (22) a	0.09623	0.4612	0.8237
Quixadá (High)	0.6667 (22) a	0.09623	0.4612	0.8237

**Table 3 viruses-14-02198-t003:** Logistic regression of resistant mosquito populations on susceptibility to transmission (saliva). Results show the means (probability scale), standard errors, and 95% confidence intervals (lower and upper means) for Zika infection of saliva. Means followed by different letters denote significant differences.

Population (Resistance Level)	Mean (No. Samples)	Std. Error of Mean	Lower Mean	Upper Mean
Rockefeller (reference strain)	0.4286 (26) a	0.09352	0.2619	0.6132
Serrinha (Low)	0.2143 (26) a	0.07754	0.09957	0.4021
Brumado (Low)	0.1212 (31) a	0.05682	0.04625	0.2818
Juazeiro do Norte (Moderate)	0.2188 (31) a	0.07308	0.1080	0.3930
Itabuna (Moderate)	0.1765 (15) a	0.09246	0.05801	0.4271
Quixadá (High)	0.2941 (15) a	0.1105	0.1280	0.5419

## Data Availability

The data presented in this study are available on request from the corresponding author.

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
