# Peer review of "Brazilian Populations of Aedes aegypti Resistant to Pyriproxyfen Exhibit Lower Susceptibility to Infection with Zika Virus"

_viruses, 2022, doi:10.3390/v14102198_

Round 1

Reviewer 1 Report

Campos et al. describes the vector competence of different populations of mosquitoes collected from five municipalities in Brazil. Studies on vector competence are always relevant from a vector biology and epidemiological stand-point. 

Major comments:

The authors should elaborate if the differences in vector competence is due to insecticide resistance or genotypes of the mosquitoes from different regions. Is it a causation or correlation? For the controls, instead of using the Rockefeller lab strain, are there mosquitoes from the same municipal that show varying resistance to PPF -> better for side by side comparison, likely that mosquitoes from same municipal have similar genetics compared to different regions. 

For the infection/dissemination rate data , did the authors perform the experiment more than once (e.g. 3 biological replicates)? Or are there any technical explanation/limitations to why the experiment was performed once. 

Line 201-204: The authors mentioned that in this study, changes in PPF resistance in mosquitoes over the 2-3 years will be investigated. However, reference 30 was cited, and the results for PPF resistance was described in the Methods (Line 240). How is the submitted manuscript different from reference 30? 

Minor comments:

Line 199-200: Contradictory as there are references 25-29 given to suggest that some studies have been done. Suggest to rephrase.

Line 372: Any reasons why there is a difference in results, e.g. enhanced susceptibility in Zika vector competence for PPF-resistance larva? 

Line 378: More elaboration why it is likely genetic hitch-hiking? Not so clear in current discussion.

Author Response

Please see the attachment which includes all responses to reviewers.

Reviewer 2 Report

This is a very well written manuscript with very important findings.  The only suggesting I have would be to explain genetic hitchhiking in more detail with more references from Culicidae if available.   

Author Response

(The authors gave the same response as above.)

Reviewer 3 Report

The paper discusses the susceptibility of Brazilian populations of Aedes aegypti resistant to 2 pyriproxyfen to infection with Zika 3 virus and is a continuation of previous studies carried out by Authors. The hypothesis of the study is clearly defined. „Introduction” section is generally well prepared. It explains to the reader the state of knowledge concerning with Zika virus. Author cite a few important references to support a statement and logically guides the reader through subsequent issues related to the topic. The results are presented in the form of tables and one figure. Data in the tables are clearly presented. The „discussion” section is conducted correctly, without being overinterpreted. References are generally relevant and referenced correctly.

Minor remarks:

1.      I suggest adding a very short information about procedures described in the WHO guidelines for larvicide bioassays and modifications introduced by Authors (mentioned on p. 3, lines 239-240).

2.      P. 7, line 346 – wrong figure number (I guess it should be denoted as 2A, not 1A).

Author Response

(The authors gave the same response as above.)

Round 2

Reviewer 1 Report

The authors have addressed my questions.